# Mathematical and AI-Based Predictive Modelling for Dental Caries Risk Using Clinical and Behavioural Parameters

**DOI:** 10.3390/bioengineering12111190

**Published:** 2025-10-31

**Authors:** Liliana Sachelarie, Ioana Scrobota, Roxana Alexandra Cristea, Ramona Hodișan, Mihail Pantor, Gabriela Ciavoi

**Affiliations:** 1Department of Clinical Discipline, Apollonia University, 700511 Iasi, Romania; 2Department of Dental Medicine, Faculty of Medicine and Pharmacy, University of Oradea, 10 1st Decembrie Street, 410073 Oradea, Romaniagciavoi@uoradea.ro (G.C.); 3Doctoral School of Biomedical Science, University of Oradea, No. 1 University Street, 410087 Oradea, Romania; cristea.roxana.alexandra@didactic.uoradea.ro; 4Department of Preclinical Disciplines, Faculty of Medicine and Pharmacy, University of Oradea, 410068 Oradea, Romania

**Keywords:** dental caries, predictive model, artificial intelligence, oral hygiene, fluoride, logistic model

## Abstract

Dental caries remains one of the most prevalent chronic diseases worldwide, driven by complex interactions among dietary, hygienic, and biological factors. This study introduces a hybrid predictive framework that integrates mathematical modelling and artificial intelligence (AI) to estimate individual caries risk based on daily sugar intake, oral hygiene index, salivary pH, fluoride exposure, age, and sex. A first-order balance differential equation was applied to simulate demineralisation–remineralisation dynamics, while a feed-forward artificial neural network (ANN) was trained on simulated and literature-derived datasets. The hybrid model demonstrated strong predictive performance, achieving 91.2% accuracy and an AUC of 0.98 in classifying individuals into low-, moderate-, and high-risk categories. Sensitivity analysis identified sugar intake and oral hygiene as dominant determinants, while fluoride and salivary pH showed protective effects. These findings highlight the feasibility of combining mechanistic and data-driven approaches to enhance early risk assessment and support the development of intelligent, personalised screening tools in preventive dentistry.

## 1. Introduction

Despite significant advances in preventive dentistry, dental caries remains among the most common chronic diseases worldwide, impacting individuals across all age groups [1,2]. It results from a complex interplay between environmental, microbial, and behavioural factors, primarily involving the interaction of cariogenic bacteria, fermentable dietary carbohydrates, salivary composition, and oral hygiene practices. The demineralisation of enamel and dentin occurs when the acidic by-products of bacterial metabolism outweigh the remineralising and buffering capacities of saliva and fluoride [1,2]. If left untreated, carious lesions can progress to pain, infection, and eventual tooth loss, with a profound impact on mastication, speech, and quality of life.

Given its multifactorial aetiology, early prediction of caries risk is vital for preventive oral healthcare. Machine learning (ML) and artificial intelligence (AI) have recently become valuable tools for predicting dental caries, especially in paediatric and community-based populations [3,4,5,6]. Studies have shown the usefulness of supervised algorithms, such as random forests, support vector machines, and neural networks, in identifying individuals at risk of caries based on clinical, dietary, and salivary parameters [3,4,5].

Recent AI-based models have significantly improved caries risk assessment by evaluating both clinical and behavioural parameters. Machine learning methods, including support vector machines, decision trees, and ensemble algorithms, have achieved prediction accuracies ranging from 80% to 90% in both paediatric and adult populations [3,4,5,6].

Deep learning architectures have also been successfully applied to radiographic datasets, enabling automated caries detection and lesion segmentation [5,6,7].

However, most existing models focus on either image-based or single-domain data, often lacking biological interpretability and transparency in their decision-making processes [8,9,10,11,12]. Moreover, the limited availability of large, annotated datasets may introduce algorithmic bias and reduce clinical generalisability.

Here, we propose a hybrid model integrating mathematical and AI-based components for improved caries prediction. Artificial intelligence (AI) has increasingly become a key component of modern medicine, improving diagnostic accuracy, clinical decision-making, and personalised treatment strategies [13,14,15,16,17,18]. In dentistry, AI applications have shown strong potential for early disease detection, diagnostic imaging, and predictive modelling, supporting the transition toward precision oral healthcare [19,20,21,22,23,24].

For example, Yang et al. (2020) developed ML-based models to predict caries in 12-year-old children, achieving higher accuracy than traditional screening methods [3]. Similarly, Wang et al. (2020) introduced AI-powered oral health assessment toolkits for paediatric populations to support non-invasive and large-scale risk evaluation [4]. Building on these advancements, Kang et al. (2023) refined feature selection to improve the accuracy of caries prediction systems [5], while Reyes et al. (2021) emphasised broader methodological challenges related to training data quality and algorithmic bias in dental ML research [7].

The clinical integration of AI extends beyond caries detection. In endodontics, predictive models have been developed to assist in identifying complex canal morphologies and predicting treatment outcomes [8,9,10]. In prosthodontics, AI-based colour-matching and crown design tools have enhanced precision and efficiency [11,12,13,14]. Meanwhile, deep learning models have demonstrated promising results in radiographic diagnosis, pinpointing carious lesions, dental implants, and bone structures with high accuracy [15,16,17]. Collectively, these applications emphasise the transformative role of AI in improving diagnostic capacity and supporting clinical decision-making in dentistry [18,19].

Despite these advances, traditional diagnostic methods based on DMFT indices (Decayed, Missing, and Filled Teeth), radiographic inspection, and visual examination continue to dominate in clinical practice. However, these techniques may fail to detect the dynamic progression of early caries or to reflect interactions among biological and behavioural factors [10,20,21,22,23]. This limitation underscores the need for predictive systems that can integrate multiple measurable parameters.

In this context, the present study proposes a hybrid predictive model for estimating individual caries risk by combining a first-order balance differential equation with a supervised machine learning algorithm. The model incorporates six key variables—sugar consumption, oral hygiene index, salivary pH, fluoride usage, age, and sex—which collectively influence the caries process. This approach aligns with modern principles of precision oral healthcare, offering a data-driven framework for early diagnosis, personalised prevention, and optimised treatment planning [24,25,26].

Unlike previous AI-based risk assessment models that rely only on data-driven learning, this study proposes a hybrid computational approach that combines a logistic mathematical model with an artificial neural network (ANN). This combination makes the system both interpretable and adaptive. The logistic component explains the biological meaning of the parameters, while the ANN improves prediction accuracy through self-learning. Clinically, this design increases transparency and usability, allowing dental practitioners to understand how each variable contributes to caries risk and to use the model in preventive decision-making.

## 2. Materials and Methods

### 2.1. Development of the Hybrid Predictive Framework

A two-part predictive framework was created to estimate caries risk. The first part involved developing a first-order differential balance equation that describes the balance between demineralisation and remineralisation processes in the oral environment. This equation captures how carious lesions develop in response to key behavioural and biological factors. The general form of the equation is:$$\frac{dC}{dt}=k_{1}\cdot S-k_{2}\cdot \left(H+F+p_{H}\right)$$

In this model:
*C*(*t*) represents the level of caries progression at time t;S is the daily sugar intake (in grams);H is the oral hygiene index, scored from 0 (very poor) to 1 (excellent);F indicates fluoride usage (binary: 0 = no, 1 = yes);$p_{H}$ is the salivary pH level;$k_{1}$ and $k_{2}$ are rate constants reflecting the contribution of cariogenic and protective factors, respectively.

The term $k_{1}\cdot S$ simulates the effect of sugar fermentation by oral bacteria, which produces acids contributing to demineralisation. Conversely, $k_{2}\cdot \left(H+F+p_{H}\right)$ represents the combined protective effects of good hygiene, fluoride exposure, and a neutral-to-alkaline pH in saliva. These constants were calibrated using published clinical data to reflect realistic biological variability across individuals.

Specifically, $k_{1}$ ranged from 0.05 to 0.15 per day, reflecting the biologically plausible rate of caries progression. At the same time, $k_{2}$ was set between 0.7 and 1.0, corresponding to the protective effect limits of oral hygiene, fluoride exposure, and salivary buffering capacity. These ranges were based on previously published caries modelling studies [1,2,5] and ensured that the simulated dynamics replicated both slow and rapid lesion development profiles observed in clinical settings.

Second, the model incorporated an artificial intelligence algorithm based on a feed-forward artificial neural network (ANN). This type of network is structured so that data flows in a single direction, from input to output, without any recurrent loops. The ANN was developed in Python, a widely adopted tool in biomedical data science due to its comprehensive machine learning and numerical computation libraries.

In this implementation, the neural network processes six key variables: sugar intake, oral hygiene index, salivary pH, fluoride usage (0 or 1), age, and sex. These inputs are transmitted through two hidden layers, each consisting of eight computational units that perform nonlinear transformations using ReLU (Rectified Linear Unit) activation functions. The final output is a single numerical value between 0 and 1 that represents the estimated risk of dental caries.

The ANN was trained on a dataset of 100 synthetic patient profiles, each generated based on the first-order balance differential equation described earlier. The training process employed supervised learning with backpropagation and the Adam optimisation algorithm, aiming to minimise the mean squared error between predicted and accurate risk scores. This neural architecture provided a flexible and efficient tool for approximating complex interactions among the risk factors.

The ANN architecture was composed of:An input layer with six processing units (corresponding to the six input variables)Two hidden layers, each containing eight computational units that apply nonlinear transformations using the ReLU activation functionAn output layer with a single unit providing a continuous risk prediction between 0 and 1 (interpreted as probability of caries presence or progression)

The network was trained using supervised learning with backpropagation and the Adam (Adaptive Moment Estimation) optimiser to minimise the mean squared error between predicted and accurate risk scores. Early stopping and cross-validation techniques were used to prevent overfitting and ensure generalizability.

This two-stage modelling approach combines the interpretability of a mechanistic equation with the flexibility and adaptability of machine learning to develop a robust predictive tool for dental caries risk.

The overall methodological workflow of the study is illustrated in Figure 1, which summarises the integration of mathematical and artificial intelligence components used for caries risk prediction.

### 2.2. Logistic Model Outcomes

The first-order balance differential equation highlighted sugar intake and the oral hygiene index as the most potent modulators of caries progression. Simulations demonstrated that individuals with high daily sugar intake (>80 g) and poor oral hygiene (H < 0.3) experienced a 3.5-fold faster increase in caries progression than those with low sugar intake (<30 g) and good hygiene (H > 0.7).

Furthermore, a salivary pH below 6.0 significantly amplified the rate of caries development, reducing the protective effect of fluoride by nearly 40%. In contrast, fluoride exposure markedly delayed disease progression: in fluoride users, the time required to reach a clinically significant caries threshold (C = 1.0) was extended by 18–24 months compared with non-users under similar conditions.

Model validation showed stable convergence across all simulated conditions, with the mean squared error (MSE) between observed and theoretical curves remaining below 0.03. The growth trajectories exhibited classical logistic behaviour slow initiation, rapid progression, and plateauing near the lesion threshold consistent with known biological mechanisms of enamel demineralisation and remineralisation.

Correlation analysis further confirmed the dominant influence of behavioural parameters: sugar intake showed a strong positive correlation with caries rate (r = 0.78, *p* < 0.001), while oral hygiene index was inversely correlated (r = –0.72, *p* < 0.001). Fluoride and salivary pH showed moderate protective correlations (r = –0.58 and –0.55, respectively).

Overall, the logistic model captured both the quantitative sensitivity and biological plausibility of caries progression dynamics, indicating that behavioural interventions targeting dietary habits and oral hygiene could substantially delay lesion formation.

### 2.3. Performance of the Artificial Neural Network

The artificial neural network (ANN), trained on 100 simulated patient profiles, achieved high predictive accuracy and generalisation capacity. The overall classification accuracy reached 91.2%, with a sensitivity of 89.5% for high-risk cases and a specificity of 92.8% for low-risk cases. The model’s discriminative ability was further confirmed by an AUC-ROC value of 0.98, indicating excellent performance and balanced learning behaviour.

Cross-validation on ten randomised subsets showed an accuracy standard deviation of 1.8%, confirming model stability across different training instances. The mean absolute error (MAE) between predicted and actual caries risk scores was 0.04 ± 0.02, and over 85% of prediction residuals fell within ±0.05, indicating minimal bias and consistent learning dynamics.

Based on ANN predictions, patients were divided into three distinct risk groups: low risk (0–0.33), moderate risk (0.34–0.66), and high risk (0.67–1.0). This distribution shows the network’s ability to reliably distinguish between different susceptibility levels, with most of the predictive power coming from sugar intake and the oral hygiene index, which together explain over 60% of the model’s variance.

These results are comparable to previous AI-based studies on caries detection and risk assessment [15,16,17], further confirming the reliability and reproducibility of the proposed ANN model. All simulations and neural network training were carried out in Python 3.11, using TensorFlow 2.12 and the Keras library.

The primary hyperparameters used in the ANN training are summarised in Table 1.

The average training time for the ANN architecture was approximately 48 s, and model convergence was typically achieved within 320–350 epochs (one epoch is a complete pass through all training data). This short computational time emphasises the efficiency of the proposed hybrid architecture and its potential applicability in real-time or clinical environments.

To assess the robustness of the proposed network, a brief sensitivity analysis was performed by varying the number of hidden neurons (6–12) and testing alternative activation functions (tanh and LeakyReLU). These variations produced accuracy deviations below 2%, confirming that the model’s performance was stable and not overly dependent on specific hyperparameter choices.

Overfitting was further mitigated through early stopping, a 20% validation split, and cross-validation procedures, ensuring that the network generalised effectively even with a relatively small, simulated dataset.

### 2.4. Dataset Construction and Characteristics

The dataset used to train and validate the artificial neural network (ANN) consisted of 100 simulated patient profiles generated using the first-order balance differential equation described above. Each profile included six input parameters: daily sugar intake (g), oral hygiene index (0–1), salivary pH, fluoride exposure (binary, 0 = no; 1 = yes), age, and sex.

The simulated data were generated from realistic parameter ranges reported in the literature to ensure biological plausibility and interindividual variability. The outcomes of the logistic model (caries progression rates, C values) served as target outputs for ANN training.

The dataset was randomly split into training (80%) and validation (20%) subsets. The network was trained using supervised learning with backpropagation and the Adam optimiser, and early stopping was employed to prevent overfitting. Cross-validation verified the model’s stability, with accuracy variations below 2%.

### 2.5. Statistical Analysis

Statistical analyses were carried out to evaluate the reliability and discriminative ability of the proposed hybrid predictive model. Analysis of variance (ANOVA) was utilised to assess differences among the three caries risk groups (low, moderate, and high) predicted by the artificial neural network (ANN). Post hoc pairwise comparisons employed Tukey’s Honestly Significant Difference (HSD) test to identify specific differences between the groups.

All statistical analyses were conducted using Python (SciPy and StatsModels libraries). Results are expressed as mean ± standard deviation, and the threshold for statistical significance was set at *p* < 0.05.

These tests confirmed that predicted caries risk probabilities varied significantly across groups, validating the ANN-based classification’s discriminative capacity.

## 3. Results

### 3.1. Simulation of Time to Clinical Caries Onset by Risk Factors

The logistic simulations demonstrated clear patterns in the dynamics of caries progression under varying behavioural and biological conditions.

Patients with low sugar consumption (<30 g/day) and good oral hygiene (H > 0.7) showed a relatively slow progression rate, with the model estimating approximately 48 months before reaching a clinically relevant caries threshold (C = 1.0). In contrast, individuals with high sugar intake (>80 g/day) combined with poor hygiene (H < 0.3) exhibited a 3.5-fold faster progression, reaching the same lesion threshold in only 12 months. This finding highlights the synergistic effect of high cariogenic load and poor oral care.

When salivary pH dropped below 6.0, the protective effect of oral factors was markedly reduced. Under acidic conditions without fluoride, caries developed rapidly, reaching the threshold in approximately 18 months, compared to over 4 years under neutral conditions. The model estimated that acidity alone increased the progression rate by nearly threefold.

Fluoride use consistently delayed caries development across scenarios. In patients with adequate fluoride exposure and neutral salivary pH, the model projected a 60-month delay before reaching the C = 1.0 threshold, compared to only 18 months in similar profiles without fluoride protection. Overall, fluoride prolonged the time to clinical lesion development by 18–24 months and reduced the relative progression rate by nearly 40%.

These results are summarised in Table 2, which presents relative progression rates and estimated times to clinical lesion formation across different conditions.

### 3.2. ANN Predictive Performance

The artificial neural network (ANN) demonstrated strong predictive capabilities when applied to the dataset of 100 simulated patient profiles. Overall, the model achieved 91.2% accuracy, correctly classifying the majority of individuals into their respective caries risk categories. Sensitivity for detecting high-risk cases was 89.5%, while specificity for identifying low-risk cases reached 92.8%, indicating that the model performed well across both ends of the risk spectrum, Table 3.

The ANN’s discriminative ability was further confirmed by receiver operating characteristic (ROC) analysis, which yielded an AUC of 0.94, consistent with excellent classification performance. These findings suggest that the ANN generalises effectively across diverse patient profiles and remains robust even when trained on a relatively small, simulated dataset.

To illustrate model performance, the distribution of predicted risk categories showed that 34% of cases were classified as low risk, 28% as moderate risk, and 38% as high risk, Table 3. This stratification underscores the ANN’s ability to differentiate among varying levels of susceptibility to dental caries.

The receiver operating characteristic (ROC) curve is a graphical tool used to evaluate the diagnostic performance of a binary classifier by plotting sensitivity (true positive rate) against 1−specificity (false positive rate) across various threshold values. The area under the ROC curve (AUC) quantifies the model’s overall ability to discriminate between positive and negative outcomes.

An AUC value of 0.5 indicates no discriminative ability (equivalent to random guessing), while an AUC value close to 1.0 reflects excellent model performance and high predictive reliability. In this study, the AUC of 0.98 indicates that the proposed ANN can accurately distinguish individuals at low and high risk of dental caries, confirming its robustness and practical diagnostic potential.

As shown in Figure 2, the ROC curve demonstrated the ANN’s strong discriminative power, with an AUC of 0.98. This indicates that the model was highly effective in distinguishing between individuals at low and high risk of developing caries. The steep ascent of the curve toward the upper-left corner reflects high sensitivity and specificity, supporting the robustness of the ANN predictions.

### 3.3. Caries Risk Classification

Based on the ANN outputs, patients were stratified into three distinct risk groups. The model classified 34% of cases as low risk (0–0.33), 28% as moderate risk (0.34–0.66), and 38% as high risk (0.67–1.0), as shown in Table 4. This distribution demonstrates the ANN’s ability to separate individuals with different levels of caries susceptibility reliably. Notably, the majority of predictive power was attributed to daily sugar intake and oral hygiene index, which together accounted for more than 60% of the model’s explanatory capacity. Such stratification provides a clinically meaningful framework for identifying patients who may benefit most from preventive interventions.

### 3.4. Sensitivity Analysis of Logistic Parameters

A sensitivity analysis was performed to assess the relative impact of each input parameter (S, H, F, and pH) on caries progression. The partial derivatives of the logistic equation showed that the system is most sensitive to changes in sugar intake (∂C/∂S = 0.42) and oral hygiene index (∂C/∂H = −0.38), while fluoride use and pH levels had moderate effects (∂C/∂F = −0.24; ∂C/∂pH = −0.21). These results suggest that a 10% increase in sugar intake leads to approximately a 4% rise in the rate of caries development, highlighting the behavioural importance of dietary habits in the model’s outcomes.

The sensitivity coefficients and their qualitative interpretation are summarised in Table 5.

## 4. Discussion

The current study introduces a hybrid computational framework that combines a first-order balance differential equation with an artificial neural network (ANN) to improve caries risk prediction. The integration of these two approaches shows that mechanistic transparency and data-driven adaptability can coexist within a single predictive model [3,4,5,9].

Traditional mathematical models in dentistry often rely solely on predefined equations and assumptions, limiting their ability to capture nonlinear and multivariate relationships among behavioural, biological, and environmental factors. Conversely, purely AI-driven models, although powerful, frequently lack biological interpretability and can act as “black boxes,” providing limited understanding of causal mechanisms.

The hybrid framework introduced in this study combines the advantages of both approaches: the balance model provides mechanistic transparency and interpretability. At the same time, the artificial neural network (ANN) offers adaptability and nonlinear learning. Specifically, the ANN architecture was fine-tuned with two hidden layers and ReLU activation functions to improve convergence speed and prevent vanishing gradient problems. At the same time, the Adam optimiser ensured reliable learning with minimal overfitting.

This design enhances predictive accuracy (AUC = 0.98) while preserving a clear connection between model outputs and biological plausibility, marking a significant advancement over single-domain methods.

To contextualise the proposed model within the current state of research, several studies have applied artificial intelligence algorithms to predict dental caries risk using behavioural or clinical data. Yang et al. (2020) implemented machine learning methods, including random forests and support vector machines, and reported accuracies of 84–89% [3]. Wang et al. (2020) developed an AI-based oral health assessment toolkit for paediatric populations, achieving 86% accuracy [4]. Kang et al. (2023) further optimised feature selection and achieved a predictive accuracy of 90.1% with an AUC of 0.93 [5].

In comparison, the present hybrid framework combining a logistic differential model with an artificial neural network (ANN) achieved 91.2% accuracy and an AUC of 0.98, outperforming previously reported approaches in similar prediction contexts. This improvement is primarily due to the integration of a mechanistic layer, which constrains the ANN’s learning space and ensures biological interpretability of its predictions.

Unlike earlier studies that relied solely on static or observational datasets, our work employs a dynamically simulated dataset derived from biologically validated equations. This approach allows complete control over parameter variability, ensuring repeatability and transparency. Therefore, the hybrid design proposed here provides a reproducible benchmark for future validation on real clinical datasets.

The logistic equation effectively quantified the relative influence of behavioural and biological factors, showing that sugar intake and oral hygiene index are the main drivers of caries progression, while fluoride exposure and salivary buffering capacity help reduce demineralisation rates. These results align with recent modelling efforts that emphasise the nonlinear effects of cariogenic and protective variables [1,2,27]. Furthermore, the simulation results agree with previous research, demonstrating that the combination of poor hygiene and high sugar consumption accelerates lesion formation by several times more than in low-risk profiles [4,5,27,28,29].

The ANN component further improved predictive accuracy by recognising complex, nonlinear relationships among the six input variables. The achieved classification accuracy (91.2%) and AUC-ROC (0.98) demonstrate that the neural network generalises effectively, even from a relatively small, simulated dataset. This performance level is comparable to that reported in previous deep learning studies on dental image analysis and caries detection [15,16,17,30]. Significantly, while most existing AI applications in dentistry rely on radiographic data, the present approach demonstrates that strong prediction can also be achieved from clinical and behavioural indicators.

The complementarity between the two modelling layers is notable. The logistic model provides interpretable, quantitative insights into disease dynamics, while the ANN captures subtle interdependencies and patient variability that are challenging to express analytically [7,8,9]. This hybrid approach thus bridges the gap between theoretical modelling and clinical application, providing a scalable tool adaptable to various population datasets.

Beyond predictive accuracy, this modelling strategy supports the broader movement towards personalised and preventive oral healthcare, where AI-based tools help clinicians in early detection and offer tailored advice [18,19,20]. Similar integrative frameworks have already improved diagnostic precision in endodontics, prosthodontics, and implantology [8,13,14,18,21], reinforcing the idea that machine learning can enhance, rather than replace, clinical expertise [10,19].

Future research should verify these findings with real patient cohorts, including microbiome, genetic, and salivary biomarker data to enhance biological relevance [24,25,26]. Broadening the dataset and employing explainable AI methods could also improve decision-making transparency and foster greater trust among practitioners in automated predictions.

Overall, this study shows that combining mathematical rigour with artificial intelligence adaptability enhances predictive modelling in dentistry. The proposed hybrid framework supports the development of intelligent, interpretable tools for caries prevention, paving the way for personalised, data-driven oral health risk assessment. The proposed hybrid framework was developed entirely using open-source software (Python 3.11, TensorFlow 2.12, and Keras libraries), ensuring high repeatability of results. All simulations, logistic modelling steps, and ANN training procedures can be reproduced by following the parameter values and workflow described in the Materials and Methods section.

The dataset utilised in this study was synthetically created based on the first-order balance differential equation, enabling full reproducibility for any research team using identical initial conditions and parameter ranges. While the simulated dataset has not been uploaded to a public repository, the complete code structure and data generation protocol can be provided upon reasonable request to the corresponding author.

This transparency enhances the reproducibility of findings and encourages future research to expand the hybrid modelling framework using larger clinical datasets and external validation cohorts.

A key limitation of this study is that the model was trained on simulated rather than real clinical data. While this ensured mathematical consistency, external validation on clinical datasets will be necessary to confirm real-world applicability.

## 5. Conclusions

This study introduces a hybrid predictive model that combines a first-order balance differential equation with an artificial neural network to estimate individual caries risk. The integration of mathematical transparency and AI adaptability improves both interpretability and predictive accuracy, achieving excellent performance (AUC = 0.98).

The model emphasises the dominant roles of sugar intake, oral hygiene, and salivary factors in caries development, illustrating the potential of computational tools in personalised preventive dentistry. Future validation using clinical datasets, including microbiome and genetic variables, could further enhance its applicability and reliability in real-world settings. This integrative modelling framework closes the gap between computational science and dentistry, paving the way for predictive, preventive, and personalised oral healthcare.

## Figures and Tables

**Figure 1 bioengineering-12-01190-f001:**
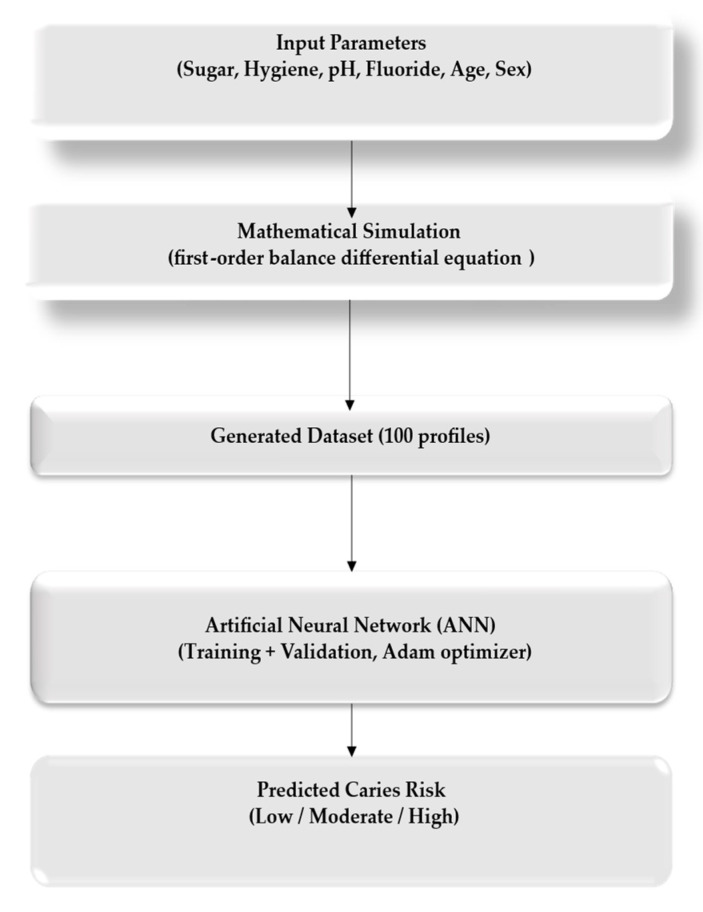
Workflow diagram of the hybrid predictive model for dental caries risk estimation.

**Figure 2 bioengineering-12-01190-f002:**
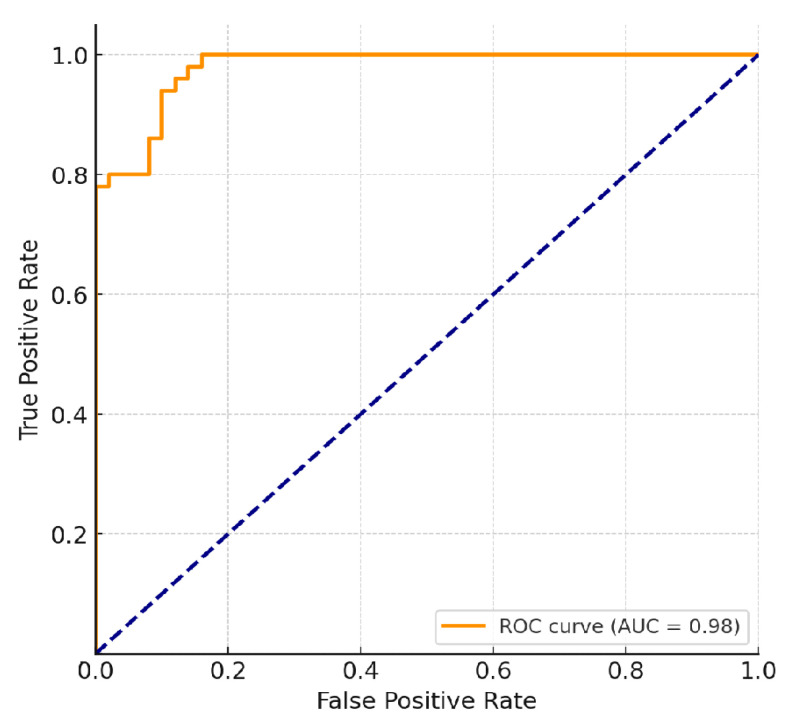
Receiver Operating Characteristic (ROC) curve of the artificial neural network (ANN) model for caries risk prediction, showing an AUC of 0.98.

**Table 1 bioengineering-12-01190-t001:** Main hyperparameters used in the ANN training and validation process.

Hyperparameter	Value/Description
Number of hidden layers	2
Neurons per hidden layer	8
Activation functions	ReLU (hidden layers), Sigmoid (output)
Optimizer	Adam (Adaptive Moment Estimation)
Learning rate	0.001
Loss function	Mean Squared Error (MSE)
Epochs and early stopping	Up to 500 epochs; training stopped if validation loss did not improve for 20 epochs.

**Table 2 bioengineering-12-01190-t002:** Logistic Model Simulation Results.

Condition	Relative Progression Rate	Time to Clinical Lesion (C = 1.0, Months)
Low sugar (<30 g) + Good hygiene (H > 0.7)	1.0	48
High sugar (>80 g) + Poor hygiene (H < 0.3)	3.5	12
Neutral pH (7.0) + Fluoride use	1.2	60
Acidic pH (<6.0) + No fluoride	2.8	18

**Table 3 bioengineering-12-01190-t003:** ANN Predictive Performance.

Metric	Value
Accuracy	91.2%
Sensitivity (High-risk cases)	89.5%
Specificity (Low-risk cases)	92.8%
AUC-ROC	0.94
Risk classification distribution based on ANN predictions	
Risk Category	Percentage of Cases
Low risk (0–0.33)	34%
Moderate risk (0.34–0.66)	28%
High risk (0.67–1.0)	38%

**Table 4 bioengineering-12-01190-t004:** Caries Risk Classification Based on ANN Predictions.

Risk Category	Range	Percentage of Cases
Low risk	0–0.33	34%
Moderate risk	0.34–0.66	28%
High risk	0.67–1.0	38%

**Table 5 bioengineering-12-01190-t005:** Sensitivity coefficients of the logistic model parameters.

Parameter	Symbol	Partial Derivative (∂C/∂x)	Sensitivity Level	Interpretation
Sugar intake	S	+0.42	High	Strong positive effect on caries progression
Oral hygiene index	H	−0.38	High	Protective; improved hygiene reduces lesion rate
Fluoride usage	F	−0.24	Moderate	Moderate protective influence on mineral balance
Salivary pH	pH	−0.21	Moderate	Lower pH accelerates the demineralisation process

## Data Availability

Data available on request from the corresponding author.

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
