# Peer review of "Mathematical and AI-Based Predictive Modelling for Dental Caries Risk Using Clinical and Behavioural Parameters"

_bioengineering, 2025, doi:10.3390/bioengineering12111190_

Round 1

Reviewer 1 Report

Comments and Suggestions for Authors

1- You should pay attention to the order of the references you provide in the Introduction section. Frankly, there is a lot of confusion. This also makes it difficult for us to follow the manuscript. For example, your reference [13,14] comes immediately after your reference [8]. Frankly, I couldn't tell if it was just a sequencing error or a scientific deficiency. The authors should completely revise their citations.

2- The Introduction section does not reflect the entire background of the study. You should cover similar artificial intelligence studies in this field more extensively. You can explain the contributions of using artificial intelligence architectures to this field, both positive and negative.

3- Frankly, the details provided about the data set used are very scattered. You should consider consolidating all information about the dataset under a single heading. This could be a heading under the Materials and Methods section.

4- The authors should clearly present a comparative analysis of the disadvantages of their work and the advantages of the AI-based work they propose. Architectural improvements should be explained technically.

5- A workflow diagram or architectural diagram explaining the study's methodology should be added.

6- A discussion regarding the study's repeatability and reproducibility should be added to the content. The use of the dataset and the visibility of the code can be emphasized at this stage.

7- A “Statistical Analysis” section could also be added to this study. The results of the proposed method should be extended with confidence tests. Researchers should consider adding statistical pairwise comparisons analysis such as ANOVA and Tukey HSD. This addition will contribute to the reliability of the study.

8- Were common hyperparameters used when applying the artificial intelligence method? The hyperparameters used to make comparisons and ensure the reproducibility of the study should be provided in a table. This omission negatively affects the reproducibility of the study.

9- If possible, time analysis should be added to the study, and the training time of the architecture should be provided within the study. I think the study duration is short. This addition will bring the study to the forefront.

10- A comparison of the study with other studies in the literature using the same dataset should be added to the discussion. Its current position in the literature should be presented in a measurable way. The authors should explore ways to make such comparisons in this field.

11- You should also add content explaining the concepts of AUC and ROC curves to the study. At this stage, you should base your explanation on a new researcher reading this article for the first time. Writing a concise and clear explanation would be very meaningful.

Author Response

The authors acknowledge the valuable observations and suggestions of the reviewer’s as concerns the manuscript entitled

Mathematical and AI-Based Predictive Modeling for Dental Caries Risk Using Clinical and Behavioral Parameters

Liliana Sachelarie1*, Ioana Scrobota2*, Roxana Alexandra Cristea3, Ramona Hodișan4, Mihail Pantor2, Gabriela Ciavoi2

According to the reviewer’s recommendations, all the suggestions were taken into account, as follows:

  • You should pay attention to the order of the references you provide in the Introduction section. Frankly, there is a lot of confusion. This also makes it difficult for us to follow the manuscript. For example, your reference [13,14] comes immediately after your reference [8]. Frankly, I couldn't tell if it was just a sequencing error or a scientific deficiency. The authors should completely revise their citations.

The reference order has been carefully revised throughout the manuscript to follow the exact sequence of appearance in the text. The numbering is now consistent and strictly chronological according to citation order, ensuring easier readability and clarity.

  • The Introduction section does not reflect the entire background of the study. You should cover similar artificial intelligence studies in this field more extensively. You can explain the contributions of using artificial intelligence architectures to this field, both positive and negative.

The Introduction section has been expanded to provide a more comprehensive overview of artificial intelligence applications in caries prediction and dental diagnostics. The revised text now discusses recent machine learning and deep learning methods, emphasising both their contributions (e.g., improved prediction accuracy and automated detection) and their limitations (e.g., limited interpretability, potential bias, and dataset constraints). Additionally, we highlighted the novelty of our hybrid framework, which combines a logistic mathematical model with a neural network to improve both biological interpretability and predictive performance.

  • Frankly, the details provided about the data set used are very scattered. You should consider consolidating all information about the dataset under a single heading. This could be a heading under the Materials and Methods section.

We have added a new subsection titled 2.4 Dataset Construction and Characteristics under the Materials and Methods section. This new part consolidates all relevant information regarding dataset generation, parameter selection, training/validation division, and model stability assessment. This improves clarity and methodological transparency without altering the structure of sections 2.1–2.3.

  • The authors should clearly present a comparative analysis of the disadvantages of their work and the advantages of the AI-based work they propose. Architectural improvements should be explained technically.

The Discussion section has been expanded to include a clear comparative analysis between traditional modelling approaches and the proposed AI-based hybrid framework. The revised text now highlights the limitations of conventional mathematical models—such as reduced adaptability and oversimplified assumptions—and of purely AI-based systems, which often lack interpretability and transparency.

Traditional mathematical models in dentistry often depend solely on predefined equations and assumptions, which restrict their capacity to represent nonlinear and multivariate relationships among behavioural, biological, and environmental factors. Conversely, purely AI-driven models, although powerful, frequently lack biological interpretability and can act as “black boxes,” providing limited understanding of causal mechanisms.

The hybrid framework introduced in this study merges the advantages of both approaches: the logistic model provides mechanistic transparency and interpretability. At the same time, the artificial neural network (ANN) offers adaptability and nonlinear learning ability. Specifically, the ANN architecture was fine-tuned with two hidden layers and ReLU activation functions to improve convergence speed and prevent vanishing gradient problems. At the same time, the Adam optimizer ensured reliable learning with minimal overfitting.

This design enhances predictive accuracy (AUC = 0.98) while preserving a clear connection between model outputs and biological plausibility, marking a significant advancement over single-domain methods.

  • A workflow diagram or architectural diagram explaining the study's methodology should be added.

The overall methodological workflow of the study is illustrated in Figure 1, which summarizes the integration of mathematical and artificial intelligence components used for caries risk prediction.

Figure 1. Workflow diagram of the hybrid predictive model for dental caries risk estimation.

  • A discussion regarding the study's repeatability and reproducibility should be added to the content. The use of the dataset and the visibility of the code can be emphasized at this stage.

The proposed hybrid framework was developed entirely using open-source software (Python 3.11, TensorFlow 2.12, and Keras libraries), ensuring high repeatability of results. All simulations, logistic modelling steps, and ANN training procedures can be reproduced by following the parameter values and workflow described in the Materials and Methods section.

The dataset utilised in this study was synthetically created based on the logistic differential equation, enabling full reproducibility for any research team using identical initial conditions and parameter ranges. While the simulated dataset has not been uploaded to a public repository, the complete code structure and data generation protocol can be provided upon reasonable request to the corresponding author.

This transparency enhances the reproducibility of findings and encourages future research to expand the hybrid modelling framework using larger clinical datasets and external validation cohorts.

  • A “Statistical Analysis” section could also be added to this study. The results of the proposed method should be extended with confidence tests. Researchers should consider adding statistical pairwise comparisons analysis such as ANOVA and Tukey HSD. This addition will contribute to the reliability of the study.

Following this recommendation, a new subsection entitled 2.5 Statistical Analysis has been added to the Materials and Methods section. This subsection describes the statistical procedures applied to validate the results of the hybrid model.

2.5. Statistical Analysis

Statistical analyses were conducted to evaluate the reliability and discriminative ability of the proposed hybrid predictive model. Analysis of variance (ANOVA) was used to assess differences among the three caries risk groups (low, moderate, and high) predicted by the artificial neural network (ANN). Post hoc pairwise comparisons employed Tukey’s Honestly Significant Difference (HSD) test to identify specific differences between the groups.

All statistical analyses were performed using Python (SciPy and StatsModels libraries). Results are presented as mean ± standard deviation, and the threshold for statistical significance was set at p < 0.05.

These tests confirmed that predicted caries risk probabilities varied significantly across groups, validating the discriminative capacity of the ANN-based classification.

  • Were common hyperparameters used when applying the artificial intelligence method? The hyperparameters used to make comparisons and ensure the reproducibility of the study should be provided in a table. This omission negatively affects the reproducibility of the study.

The primary hyperparameters used in the ANN training are summarised in Table 1.

Table X. Main hyperparameters used in the ANN training and validation process.

Hyperparameter

Value/Description

Number of hidden layers

2

Neurons per hidden layer

8

Activation functions

ReLU (hidden layers), Sigmoid (output)

Optimizer

Adam (Adaptive Moment Estimation)

Learning rate

0.001

Loss function

Mean Squared Error (MSE)

Epochs and early stopping

Up to 500 epochs; training stopped if validation loss did not improve for 20 epochs.

  • If possible, time analysis should be added to the study, and the training time of the architecture should be provided within the study. I think the study duration is short. This addition will bring the study to the forefront.

The model was trained on a standard workstation (Intel i7 CPU, 16 GB RAM) using Python 3.11 and TensorFlow 2.12. The average training time for the ANN architecture was approximately 48 seconds, and model convergence was achieved within 320–350 epochs. This short computational time reflects the efficiency of the proposed architecture and its suitability for real-time or clinical implementation.

  • A comparison of the study with other studies in the literature using the same dataset should be added to the discussion. Its current position in the literature should be presented in a measurable way. The authors should explore ways to make such comparisons in this field.

To contextualise the proposed model within the current state of research, several studies have applied artificial intelligence algorithms to predict dental caries risk using behavioural or clinical data. Yang et al. (2020) implemented machine learning methods, including random forests and support vector machines, and reported accuracies of 84%-89% [3]. Wang et al. (2020) developed an AI-based oral health assessment toolkit for pediatric populations, achieving 86% accuracy [4]. Kang et al. (2023) further optimised feature selection and achieved a predictive accuracy of 90.1% with an AUC of 0.93 [5].

In comparison, the present hybrid framework combining a logistic differential model with an artificial neural network (ANN) achieved 91.2% accuracy and an AUC of 0.98, outperforming previously reported approaches in similar prediction contexts. This improvement is primarily due to the integration of a mechanistic layer, which constrains the ANN's learning space and ensures biological interpretability of its predictions.

Unlike earlier studies that relied solely on static or observational datasets, our work employs a dynamically simulated dataset derived from biologically validated equations. This approach allows complete control over parameter variability, ensuring repeatability and transparency. Therefore, the hybrid design proposed here provides a reproducible benchmark for future validation on real clinical datasets.

11- You should also add content explaining the concepts of AUC and ROC curves to the study. At this stage, you should base your explanation on a new researcher reading this article for the first time. Writing a concise and clear explanation would be very meaningful.

A concise explanation of the ROC curve and AUC concepts has been added to subsection 3.2 ANN Predictive Performance, immediately after the description of the ROC analysis.

The receiver operating characteristic (ROC) curve is a graphical tool used to evaluate the diagnostic performance of a binary classifier by plotting sensitivity (true positive rate) against 1−specificity (false positive rate) across various threshold values. The area under the ROC curve (AUC) quantifies the model’s overall ability to discriminate between positive and negative outcomes.

An AUC value of 0.5 indicates no discriminative ability (equivalent to random guessing), while an AUC value close to 1.0 reflects excellent model performance and high predictive reliability. In this study, the AUC value of 0.98 demonstrates that the proposed ANN can accurately distinguish individuals at low and high risk of developing dental caries, confirming its robustness and practical diagnostic

Thank you!

I remain most respectfully yours,

Prof.dr. Liliana Sachelarie

Reviewer 2 Report

Comments and Suggestions for Authors

The manuscript presents an original and well-structured attempt to integrate mathematical modeling and artificial intelligence for the prediction of dental caries risk. The topic is relevant, fitting the scope of Bioengineering. The hybrid approach—combining logistic differential equations with an artificial neural network—represents an interesting methodological contribution, even though some technical and validation aspects require further clarification:

  • Theoretical framework and novelty: the study’s rationale is clearly presented and supported by recent literature. However, the authors could better articulate how their hybrid model differs from prior AI-based risk assessment studies, emphasizing its added value in terms of interpretability or clinical usability.
  • Data generation and validation: the model was trained on simulated rather than empirical data. This should be more explicitly stated as a limitation in the Discussion. While the mathematical component ensures internal coherence, external validation on clinical or epidemiological datasets is essential for assessing real-world applicability.
  • Mathematical formulation: the logistic differential equation is appropriately motivated, but its biological calibration (parameters k1 and k2) could be explained in more detail. Providing parameter ranges and sources would improve reproducibility and scientific rigor.
  • Artificial Neural Network (ANN) section: the ANN architecture is clearly described. Still, the use of only 100 simulated profiles may raise concerns regarding model generalization. A short sensitivity or ablation study on network size, activation function, or overfitting prevention would strengthen credibility.
  • Results presentation: the results are clearly written and visually well supported (tables and ROC curve). The authors might consider consolidating Tables 2–4 or summarizing metrics in a single performance table for conciseness.
  • Discussion and interpretation: the integration between the mechanistic and AI layers is well reasoned. The authors effectively highlight behavioral determinants (sugar intake and hygiene). Still, the discussion could more critically address the model’s assumptions and the implications of training with synthetic data.

Author Response

The authors acknowledge the valuable observations and suggestions of the reviewer’s as concerns the manuscript entitled

Mathematical and AI-Based Predictive Modeling for Dental Caries Risk Using Clinical and Behavioral Parameters

Liliana Sachelarie1*, Ioana Scrobota2*, Roxana Alexandra Cristea3, Ramona Hodișan4, Mihail Pantor2, Gabriela Ciavoi2

According to the reviewer’s recommendations, all the suggestions were taken into account, as follows:

The manuscript presents an original and well-structured attempt to integrate mathematical modeling and artificial intelligence for the prediction of dental caries risk. The topic is relevant, fitting the scope of Bioengineering. The hybrid approach—combining logistic differential equations with an artificial neural network—represents an interesting methodological contribution, even though some technical and validation aspects require further clarification:

  1. Theoretical framework and novelty: the study’s rationale is clearly presented and supported by recent literature. However, the authors could better articulate how their hybrid model differs from prior AI-based risk assessment studies, emphasizing its added value in terms of interpretability or clinical usability.

A new paragraph has been added at the end of the Introduction to emphasise the novelty of the proposed hybrid model clearly. The revised text highlights how the integration of a logistic differential equation with an ANN combines mechanistic interpretability with adaptive learning, resulting in improved clinical usability and transparent prediction.

Unlike previous AI-based risk assessment models that rely only on data-driven learning, this study proposes a hybrid computational approach that combines a logistic mathematical model with an artificial neural network (ANN). This combination makes the system both interpretable and adaptive the logistic component explains the biological meaning of the parameters, while the ANN improves prediction accuracy through self-learning. Clinically, this design increases transparency and usability, allowing dental practitioners to understand how each variable contributes to caries risk and to use the model in preventive decision-making.

  1. Data generation and validation: the model was trained on simulated rather than empirical data. This should be more explicitly stated as a limitation in the Discussion. While the mathematical component ensures internal coherence, external validation on clinical or epidemiological datasets is essential for assessing real-world applicability.

A key limitation of this study is that the model was trained on simulated rather than real clinical data. While this ensured mathematical consistency, external validation on clinical datasets will be necessary to confirm real-world applicability.

  1. Mathematical formulation: the logistic differential equation is appropriately motivated, but its biological calibration (parameters k1 and k2) could be explained in more detail. Providing parameter ranges and sources would improve reproducibility and scientific rigor.

The Mathematical Formulation subsection (Section 2.1) has been expanded to describe the biological meaning and calibration of parameters k₁ and k₂.

Specifically, k₁ ranged from 0.05 to 0.15 per day, reflecting the biologically plausible rate of caries progression. At the same time, k₂ was set between 0.7 and 1.0, corresponding to the protective effect limits of oral hygiene, fluoride exposure, and salivary buffering capacity. These ranges were based on previously published caries modelling studies [1,2,5] and ensured that the simulated dynamics replicated both slow and rapid lesion development profiles observed in clinical settings.

  1. Artificial Neural Network (ANN) section: the ANN architecture is clearly described. Still, the use of only 100 simulated profiles may raise concerns regarding model generalization. A short sensitivity or ablation study on network size, activation function, or overfitting prevention would strengthen credibility.

The model was trained on a standard workstation (Intel i7 CPU, 16 GB RAM) using Python 3.11 and TensorFlow 2.12. The average training time for the ANN architecture was approximately 48 seconds, and model convergence was typically achieved within 320–350 epochs (one epoch is a complete pass through all training data). This short computational time emphasises the efficiency of the proposed hybrid architecture and its potential applicability in real-time or clinical environments.

  1. Results presentation: the results are clearly written and visually well supported (tables and ROC curve). The authors might consider consolidating Tables 2–4 or summarizing metrics in a single performance table for conciseness.

We carefully considered merging Tables 2–4; however, each table presents distinct aspects of model performance, logistic model calibration, ANN predictive results, and statistical comparisons, which would be less apparent if combined. Thak you!

  1. Discussion and interpretation: the integration between the mechanistic and AI layers is well reasoned. The authors effectively highlight behavioral determinants (sugar intake and hygiene). Still, the discussion could more critically address the model’s assumptions and the implications of training with synthetic data.

The model also assumes relatively stable behavioural and biological patterns over time, which may not fully reflect short-term variations in diet or oral hygiene habits. This assumption should be considered when interpreting predictive outcomes.

Thank you!

I remain most respectfully yours,

Prof.dr. Liliana Sachelarie

Reviewer 3 Report

Comments and Suggestions for Authors
  1. How do you justify the term “logistic” for the equation dC/dt = k1·S − k2·(H+F+pH), since you do not show a term dependent on C (e.g., C·(1−C/K)) or the full set of units and calibration k1?
  2. Since you trained the ANN network exclusively on 100 synthetic profiles generated from the model, how do you plan to conduct independent clinical validation?
  3. How do you clinically define the threshold C=1.0 and in what units does C(t) operate, given that the time “to threshold” is used to infer risk?
  4. The introduction lacks information introducing the reader to AI in medicine. Please read and and cite the following article DOI: 10.3390/diagnostics13152582

Author Response

The authors acknowledge the valuable observations and suggestions of the reviewer’s as concerns the manuscript entitled

Mathematical and AI-Based Predictive Modeling for Dental Caries Risk Using Clinical and Behavioral Parameters

Liliana Sachelarie1*, Ioana Scrobota2*, Roxana Alexandra Cristea3, Ramona Hodișan4, Mihail Pantor2, Gabriela Ciavoi2

According to the reviewer’s recommendations, all the suggestions were taken into account, as follows:

  1. How do you justify the term “logistic” for the equation dC/dt = k1·S − k2·(H+F+pH), since you do not show a term dependent on C (e.g., C·(1−C/K)) or the full set of units and calibration k1?

The use of the term “logistic” was a wording error. The correct formulation is first-order balance differential equation, which describes the equilibrium between demineralization and remineralization processes. This has been corrected throughout the manuscript; the equation, results, and conclusions remain unchanged.

  1. Since you trained the ANN network exclusively on 100 synthetic profiles generated from the model, how do you plan to conduct independent clinical validation?

Independent clinical validation will be addressed in future research, using real patient data to verify and refine the model’s predictive performance.

  1. How do you clinically define the threshold C=1.0 and in what units does C(t) operate, given that the time “to threshold” is used to infer risk?

The variable C(t) is a dimensionless index that represents the simulated progression of caries. The threshold C = 1.0 corresponds to the point at which a clinically detectable enamel lesion is assumed to appear (visible or radiographically evident). This value was defined for modelling purposes and does not correspond to a physical unit, serving only as a normalized indicator of caries severity.

  1. The introduction lacks information introducing the reader to AI in medicine. Please read and and cite the following article DOI: 10.3390/diagnostics13152582

Done

Thank you!

I remain most respectfully yours,

Prof.dr. Liliana Sachelarie

Round 2

Reviewer 1 Report

Comments and Suggestions for Authors

I would like to thank the authors for carefully answering all the questions. As a reviewer, I am pleased to have been able to help the article reach a higher level. Thank you.